

# Calibration adjustments to address bias in mortality analyses due to informative sampling—a census-linked survey analysis in Switzerland

André Moser[1,2], Matthias Bopp[3], Marcel Zwahlen[2] and Swiss National Cohort study group

[1] Department of Geriatrics, Inselspital, University Hospital, and University of Bern, Bern, Switzerland
[2] Institute of Social and Preventive Medicine, University of Bern, Bern, Switzerland
[3] Epidemiology, Biostatistics and Prevention Institute, University of Zürich, Zürich, Switzerland

Corresponding author
André Moser,
andre.moser@ispm.unibe.ch

## ABSTRACT

**Background**. Sampling bias, like survey participants' nonresponse, needs to be adequately addressed in the analysis of sampling designs. Often survey weights will be calibrated on specific covariates related to the probability of selection and nonresponse to get representative population estimates. However, such calibrated survey (CS) weights are usually constructed for cross-sectional results, but not for longitudinal analyses. For example, when the outcome of interest is time to death, and sampling selection is related to time to death and censoring, sampling is informative. Then, unweighted or CS weighted inferential statistical analyses may be biased. In 2010, Switzerland changed from a decennial full enumeration census to a yearly registry-based (i.e., data from harmonised community registries) and a survey-based census system. In the present study, we investigated the potential bias due to informative sampling when time to death is the outcome of interest, using data from the new Swiss census system.

**Methods**. We analysed more than 6.5 million individuals aged 15 years or older from registry-based census data from years 2010 to 2013, linked with mortality records up to end of 2014. Out of this population, a target sample of 3.5% was sampled from the Swiss Federal Statistical Office (SFSO) in a stratified yearly micro census. The SFSO calculated CS weights to enable representative population estimates from the micro census. We additionally constructed inverse probability (IP) weights, where we used survival information in addition to known sampling covariates. We compared CS and IP weighted mortality rates (MR) and life expectancy (LE) with estimates from the underlying population. Additionally, we performed a simulation study under different sampling and nonresponse scenarios.

**Results**. We found that individuals who died in 2011, had a 0.67 (95% CI [0.64–0.70]) times lower odds of participating in the 2010 micro census, using a multivariable logistic regression model with covariates age, gender, nationality, civil status, region and survival information. IP weighted MR were comparable to estimates from the total population, whereas CS weighted MR underestimated the population MR in general. The IP weighted LE estimates at age 30 years for men were 50.9 years (95% CI [50.2–51.6] years), whereas the CS weighted overestimated LE by 2.5 years. Our results from

the simulation study confirmed that IP weighted models are comparable to population estimates.

**Conclusion**. Mortality analyses based on the new Swiss survey-based census system may be biased, because of informative sampling. We conclude that mortality analyses based on census-linked survey data have to be carefully conducted, and if possible, validated by registry information to allow for unbiased interpretation and generalisation.

## INTRODUCTION

Population-based longitudinal studies are a key tool for epidemiologists to investigate individual and ecological risk factors on disease development and mortality over time. In many countries, including the United Kingdom, Germany and the Nordic countries, nationwide cohort studies have been established by linking information from population registries and secondary data sources (*Olsen et al., 2001*; *Ahrens et al., 2014*; *Connelly & Platt, 2014*). In Switzerland, a nationwide cohort study was established in 2008, based on a probabilistic record linkage of census and mortality information covering the whole Swiss population (*Bopp et al., 2009*; *Spoerri et al., 2010*). This cohort study relied on an almost complete census (coverage of 98.6% (*Renaud, 2004*)) and allowed for almost unbiased mortality analyses (*Schmidlin et al., 2013*; *Moser et al., 2014*).

In 2010, Switzerland changed from a full enumeration census—which was conducted every 10 years—to a registry-based and survey-based census system. This new Swiss census system is based on a yearly updated registry-based census—which collects information from harmonised community registries covering the entire resident population of Switzerland (STATPOP)—and a micro census of roughly 3.5% of the resident Swiss population aged 15 years or older, living in a private household. This micro census collects individual and household information, which is not available in the STATPOP population (e.g., attained educational level). The new census system aims to be more cost-efficient and to require less administrative resources compared to the traditional full enumeration census. However, because the micro census is based on only a stratified sampled subpopulation, the sampling design needs to be addressed in descriptive and inferential statistical analyses to get valid population estimates (*Korn & Graubard, 1999*; *Fuller, 2009*).

The Swiss Federal Statistical Office (SFSO) planned and performed the sampling of the micro census, and provided calibrated survey (CS) weights for the analysis of the micro census sample. CS weights address the sampling process (i.e., probabilities of selection) and account for sampling biases (e.g., nonresponse adjustments) to get consistent statistical estimates for the whole Swiss population in weighted descriptive analyses (*Assoulin, 2012*). These analyses are valid for cross-sectional results, but not necessary for longitudinal analyses. For example, if the outcome of interest is time to death and the sampling selection is not independent of the survival time and censoring, given other covariates, the sampling

design is called ''informative'' (*Lawless, 2003*; *Boudreau & Lawless, 2006*). When sampling is informative, the outcome of interest is not missing at random and unweighted or CS weighted inferential statistical analyses may be biased. IP weighting models, which account for the probability of survey inclusion and censoring, could correct for such bias and lead to valid survival estimates (*Scharfstein & Robins, 2002*; *Little, 2004*; *Boudreau & Lawless, 2006*).

In the present study, we performed time to death analyses using data from the Swiss micro census. Specifically, we calculated mortality rates (i.e., survival time is an offset variable in a Poisson regression model) and life expectancy (i.e., survival time is directly modelled in a censored skew-normal regression model (*Moser, Clough-Gorr & Zwahlen, 2015*)). In Switzerland, non-responders of surveys have a higher risk of dying compared to responders (*Bopp, Braun & Faeh, 2014*). Thus, we hypothesised that bias due to informative sampling may exist in CS weighted analyses, caused by unmeasured predictors (e.g., health status), which are possibly related to the sampling process and mortality. We therefore compared CS weighted results with those from an IP weighting approach, which uses known sampling covariates and survival information, and validated our findings using data from the total Swiss population. Additionally, we performed a simulation study to investigate the bias due to informative sampling under different sampling and nonresponse scenarios.

## METHODS

### Data availability
We provide an anonymised dataset of the STATPOP and SE populations of the year 2010 to allow reproducibility of our main results. By law, we are not allowed to provide exact dates (date of birth and date of death). Thus, we rounded necessary main analysis variables (age and follow-up time) to one decimal place. Rounding had no influence on obtained results, compared to results using exact date information. Results in the manuscript are based on exact information. Analysis code for all results is provided in a Supplemental File.

### Human ethics
Federal laws gave the federal authorities the right to collect without consent the census and mortality data used in the SNC. Approval for the anonymous linkage in the SNC was obtained from the Ethics Committees of the Cantons of Zürich (approval no. 13/06) and Bern (approval no. 205/06).

### Registry-based population
We used the STATPOP population of the year 2010 with permanent residents (which excludes cross-border workers) in Switzerland aged 15 years or older. We used information age (5 year categories), gender, civil status (single, married, widowed, other), nationality (Swiss, European Economic Area (EEA), other Europe, other World), and regional level (cantons of Zurich (subdivided in City of Zurich and remainder), Bern (subdivided in City of Bern, Bernese Jura, and remainder), Lucerne, Uri, Schwyz, Obwalden, Nidwalden, Glarus, Zug, Fribourg, Solothurn, Basel-Stadt, Basel-Landschaft, Schaffhausen, Appenzell Ausserrhoden, Appenzell Innerrhoden, St. Gallen, Graubuenden, Aargau, Thurgau, Ticino, Vaud, Valais, Neuchâtel, Geneva, Jura).
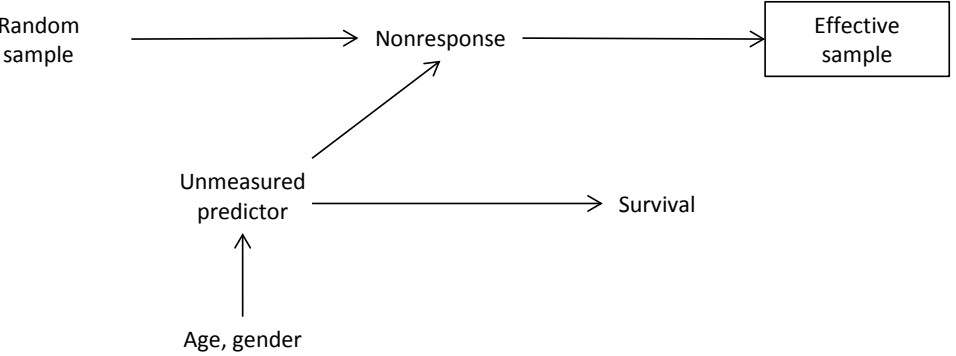

**Figure 1 Directed acyclic graph for informative sampling situation.** An unmeasured predictor, say poor health status, is directly related to survival. Older men (observed predictors age and gender) are assumed to have more likely a poor health status. Further, individuals with a poor health status are less likely to respond to a survey questionnaire (nonresponse). Nonresponse has a direct effect on the effective sample. Conditioning on the effective sample (collider, indicated by box) opens the path "Random sample -> Nonresponse -> Unmeasured predictor -> Survival", and introduces a spurious association between the sampling selection and survival.

## Micro census

The micro census of the year 2010, subsequently called 'structural enquiry' (SE), is a stratified sample of the permanent residents in Switzerland aged 15 years or older, living in a private household. The sampling aim and planned accuracy of the SE has been described elsewhere (*Assoulin, 2012*; *Qualité, Statistik & De Europe, 2014*). In brief, the SE is a regionally-stratified proportional sample with a target sampling fraction of approximately 3.5%. The regions had the opportunity to increase the sample size to increase precision of statistical estimates, which led to a final sampling fraction of 5.5% (*Rochat, Kauthen & Eichenberger, 2009*). For the SE 2010, the response rate was 87.1% (*Qualité, Statistik & De Europe, 2014*). The SFSO calculated individual CS weights based on sampling inclusion and response probabilities (*Assoulin, 2012*; *Qualité, Statistik & De Europe, 2014*).

## Mortality information

Survival status for the years 2011–2014 could be one-to-one linked to the STATPOP and the SE 2010 population by unique anonymous personal identifier. The percentage of non-linked death records is 0.2%. Vital status is ascertained by death certificates. We used information about vital status as binary variable (dead: yes or no) and survival time.

## Simulation study

We investigated the influence of informative sampling bias in a simulation study. We simulated a population of 1,000,000 individuals with different sampling and nonresponse scenarios. Our main outcomes were number of deaths and time to death to calculate MR. We performed a simulation study as depicted in Fig. 1: we assumed an unobserved predictor (e.g., health status) and observed predictors (e.g., age and gender). On a population level, there is a direct effect of age and gender on health status, e.g., older men are more likely to have a poor health status. Further, health status has a causal effect on survival. Age and

gender are related to survival through the unmeasured predictor. A random sample was drawn from the underlying population (planned sampling design). We assumed that a poor health status had a direct effect on the nonresponse, leading to a smaller effective sample. Conditioning on the effective sample opens the path between the random sample and survival (through nonresponse and the unmeasured confounder) and will lead to a spurious association in survival estimates. We assumed a constant risk of dying over time. For the effect sizes we chose an odds ratio of 4 for the association between the unmeasured confounder and nonresponse, and a 4-times higher hazard of dying for individuals with bad health, as reported in *Pizzi et al. (2011)*. We calculated CS weights from varying probabilities of sampling and nonresponse. IP weights were constructed from the probability of 'being sampled' from the total population using vital status. We performed the following simulation scenarios: the overall percentage of deaths was set to 10% and 25% for the total population, the sampling fraction was set to 1% and 10%, and the percentage of nonresponse was set to 10% and 25%. We ran 1,000 simulation replications. We calculated MR from unweighted Poisson regression models, and models using CS and IP weights, using covariates age and gender. We reported the 1,000 replicated mortality rate estimates in scatter plots with bivariate mean location centre estimates and 95% confidence ellipses from a Gaussian distribution function.

## Statistical analysis

We described the study populations by frequencies (*n*), percentages (%), mean and standard deviation (SD). We reported crude odds ratios of participation in the SE 2010 using logistic regression models, for variables age, gender, nationality, civil status and vital status. We calculated MR using Poisson regression models. Remaining LE at age 30 years was calculated from weighted censored skew-normal regression models, accounting for delayed entry (*Moser, Clough-Gorr & Zwahlen, 2015*). Because MR and LE were a priori known to differ by gender, we performed analyses for men and women, separately. We calculated IP weights using logistic regression models with an indicator of being a SE 2010 participant in the STATPOP 2010 population. Predictors were chosen according to a priori known covariates for CS weights, i.e., age, gender, nationality, civil status, region (*Assoulin, 2012*), and additional survival information. We included interactions between age, gender and survival information. Because the main outcome of interest is time to event we included the Nelson-Aalen estimator in the final IP weighting model (*White & Royston, 2009*). Observation time started December 31, 2010, and ended on date of death or December 31, 2011, whichever came first. The end date changed with the investigated time period, the latest was December 31, 2014. We calculated marginal distributions of the total population from CS and IP weighted estimates. All data were analysed with Stata 14.1 (StataCorp LP, College Station, TX, USA) or R version 3.3.1 (*R Development Core Team, 2016*). The Stata code and command for the censored skew-normal approach is available at https://github.com/MoserGitHub/censn.

## Sensitivity analysis

We investigated whether the hypothesised bias is present in the years 2011–2013. First, we reported crude odds ratios of participation in one of the SE 2011–2013, analysing covariates

age, gender, nationality, civil status and vital status (for brevity we did not report estimates of the regional covariate). Second, we investigated whether the hypothesised bias of CS weighted analysis is similarly present in the years 2011–2013, by calculating 1-year MR.

## RESULTS

### Study population

Table 1 summarises the baseline characteristics of the STATPOP population and the SE sample of 2010, by survival status. At 31 December 2010, the STATPOP population consisted of 6.7 million individuals. 49% of the population was male, 51% female. More than half of the individuals were married (51.6%) and most of the individuals were Swiss (77.3%). Until end of 2011, a total of 61,539 persons died. The 2010 SE sample consisted of 317,079 sampled individuals. The marginal distributions of the covariates for the SE sample and the STATPOP population were roughly the same, except for the regional covariate, which is a sampling stratification covariate. There were differences in individuals who died until end of 2011 in the SE sample and the STATPOP population. Individuals who died in the STATPOP population were slightly older: only 50% of the individuals were 80 years or older, compared to 57% in the STATPOP. Further, only 30% of widowed women died in the SE sample, compared to 37.5% in the STATPOP population. Table S1 summarises the marginal distributions of the baseline characteristics for CS and IP weighted estimates. The marginal distributions were comparable between the STATPOP population and the weighted estimates from the SE sample.

### Participation in SE 2010

Table 2 summarises crude odds ratios of being 2010 SE participant with the 2010 STATPOP population as underlying population, using covariates age, gender, nationality, civil status, region and vital status. Individuals aged 85 years or more were less likely to be sampled in the SE 2010 (OR 0.91, 95% CI [0.088–0.93]), compared to the youngest age group. Foreigners had a lower odds of participating than Swiss nationals (e.g., OR for persons from EEA 0.94, 95% CI [0.93–0.95]). Individuals who died in 2011 had a 0.67, 95% CI [0.64–0.70], lower odds of participating in the SE 2010.

### Mortality analysis

Figure 2 shows the estimated 1-year MR for the STATPOP population and the SE of the year 2010, by age categories and gender. STATPOP MR (red coloured lines) serve as the reference MR, because they contain full population information. IP weighted MR (green coloured lines) were comparable to STATPOP MR, especially for ages 40 years or older. Unweighted (blue coloured lines) MR and CS weighted (purple coloured lines) MR underestimated the STATPOP MR. Figure 3 shows the estimated LE at age 30 years. The male population of the STATPOP had an estimated LE of 50.9 years (95% CI [50.8–51.0] years). The IP weighted LE estimates for men were 50.9 years (95% CI [50.2–51.6] years), whereas the CS weighted were 53.4 years (95% CI [52.7–54.0] years). The results for women were comparable, i.e., CS weighted results overestimated LE, compared to IP weighted results. Table S2 describes the distribution of individuals who died in 2011 for the STATPOP and

**Table 1  Description of registry-based population (STATPOP) and the micro census (structural enquiry, SE) 2010, by survival status[a].**

| | | Registry-based population 2010 | | Structural enquiry sample 2010 | |
|---|---|---|---|---|---|
| | | Alive (*n* = 6,667,824) *n* (%)/mean (SD) | Died (*n* = 61,539) *n* (%)/mean (SD) | Alive (*n* = 315,108) *n* (%)/mean (SD) | Died (*n* = 1,971) *n* (%)/mean (SD) |
| Age categories | [15, 20) | 453,418 (6.8) | 125 (0.2) | 19,890 (6.3) | 6 (0.3) |
| | [20, 25) | 497,932 (7.5) | 195 (0.3) | 20,467 (6.5) | 10 (0.5) |
| | [25, 30) | 535,327 (8.0) | 179 (0.3) | 23,935 (7.6) | 3 (0.2) |
| | [30, 35) | 542,177 (8.1) | 250 (0.4) | 26,445 (8.4) | 8 (0.4) |
| | [35, 40) | 562,531 (8.4) | 330 (0.5) | 27,456 (8.7) | 11 (0.6) |
| | [40, 45) | 636,715 (9.5) | 662 (1.1) | 30,282 (9.6) | 21 (1.1) |
| | [45, 50) | 651,880 (9.8) | 1,087 (1.8) | 31,172 (9.9) | 42 (2.1) |
| | [50, 55) | 565,305 (8.5) | 1,525 (2.5) | 26,896 (8.5) | 52 (2.6) |
| | [55, 60) | 484,779 (7.3) | 2,048 (3.3) | 23,522 (7.5) | 70 (3.6) |
| | [60, 65) | 456,872 (6.9) | 3,110 (5.1) | 22,204 (7.0) | 122 (6.2) |
| | [65, 70) | 393,251 (5.9) | 4,259 (6.9) | 19,730 (6.3) | 180 (9.1) |
| | [70, 75) | 298,540 (4.5) | 5,082 (8.3) | 15,000 (4.8) | 163 (8.3) |
| | [75, 80) | 248,981 (3.7) | 7,565 (12.3) | 12,520 (4.0) | 300 (15.2) |
| | [80, 85) | 184,460 (2.8) | 10,711 (17.4) | 9,040 (2.9) | 323 (16.4) |
| | ≥85 years | 155,656 (2.3) | 24,411 (39.7) | 6,549 (2.1) | 660 (33.5) |
| Gender | Men | 3,270,394 (49.0) | 29,804 (48.4) | 151,684 (48.1) | 1,099 (55.8) |
| | Women | 3,397,430 (51.0) | 31,735 (51.6) | 163,424 (51.9) | 872 (44.2) |
| Nationality | Swiss | 5,147,117 (77.2) | 56,418 (91.7) | 249,184 (79.1) | 1,779 (90.3) |
| | EEA | 989,417 (14.8) | 4,309 (7.0) | 45,200 (14.3) | 172 (8.7) |
| | Other Europe | 317,233 (4.8) | 538 (0.9) | 12,503 (4.0) | 14 (0.7) |
| | Other World | 214,057 (3.2) | 274 (0.4) | 8,221 (2.6) | 6 (0.3) |
| Civil status | Single | 2,224,776 (33.4) | 7,455 (12.1) | 101,868 (32.3) | 189 (9.6) |
| | Married | 3,449,816 (51.7) | 24,937 (40.5) | 166,452 (52.8) | 999 (50.7) |
| | Widowed | 387,352 (5.8) | 23,109 (37.5) | 17,942 (5.7) | 592 (30.0) |
| | Other | 605,880 (9.1) | 6,038 (9.8) | 28,846 (9.2) | 191 (9.7) |
| Region | City of Zurich | 334,727 (5.0) | 3,438 (5.6) | 39,120 (12.4) | 286 (14.5) |
| | Remainder of canton Zurich | 837,660 (12.6) | 6,681 (10.9) | 24,948 (7.9) | 119 (6.0) |
| | City of Bern | 113,968 (1.7) | 1,331 (2.2) | 13,139 (4.2) | 111 (5.6) |
| | Bernese Jura | 43,374 (0.7) | 518 (0.8) | 2,899 (0.9) | 22 (1.1) |
| | Remainder of canton Bern | 689,868 (10.3) | 7,072 (11.5) | 21,957 (7.0) | 130 (6.6) |
| | Lucerne | 318,109 (4.8) | 2,750 (4.5) | 19,208 (6.1) | 96 (4.9) |
| | Uri | 29,375 (0.4) | 315 (0.5) | 976 (0.3) | 5 (0.3) |
| | Schwyz | 122,585 (1.8) | 1,053 (1.7) | 3,589 (1.1) | 19 (1.0) |
| | Obwalden | 29,675 (0.4) | 249 (0.4) | 828 (0.3) | 4 (0.2) |
| | Nidwalden | 34,471 (0.5) | 270 (0.4) | 1,039 (0.3) | 6 (0.3) |
| | Glarus | 32,785 (0.5) | 364 (0.6) | 914 (0.3) | 7 (0.4) |
| | Zug | 94,883 (1.4) | 704 (1.1) | 5,205 (1.7) | 23 (1.2) |
| | Fribourg | 228,838 (3.4) | 1,948 (3.2) | 6,570 (2.1) | 39 (2.0) |
| | Solothurn | 216,854 (3.3) | 2,161 (3.5) | 6,598 (2.1) | 54 (2.7) |

**Table 1** (*continued*)

| | Registry-based population 2010 | | Structural enquiry sample 2010 | |
| | Alive ($n = 6,667,824$) $n$ (%)/mean (SD) | Died ($n = 61,539$) $n$ (%)/mean (SD) | Alive ($n = 315,108$) $n$ (%)/mean (SD) | Died ($n = 1,971$) $n$ (%)/mean (SD) |
|---|---|---|---|---|
| Basel-Stadt | 162,021 (2.4) | 2,026 (3.3) | 5,113 (1.6) | 26 (1.3) |
| Basel-Landschaft | 234,029 (3.5) | 2,195 (3.6) | 7,366 (2.3) | 44 (2.2) |
| Schaffhausen | 65,688 (1.0) | 713 (1.2) | 2,001 (0.6) | 18 (0.9) |
| Appenzell Ausserrhoden | 43,940 (0.7) | 444 (0.7) | 1,382 (0.4) | 15 (0.8) |
| Appenzell Innerrhoden | 12,738 (0.2) | 135 (0.2) | 381 (0.1) | 2 (0.1) |
| St. Gallen | 405,528 (6.1) | 3,657 (5.9) | 12,174 (3.9) | 57 (2.9) |
| Graubuenden | 174,072 (2.6) | 1,615 (2.6) | 5,070 (1.6) | 32 (1.6) |
| Aargau | 513,802 (7.7) | 4,350 (7.1) | 30,046 (9.5) | 164 (8.3) |
| Thurgau | 209,311 (3.1) | 1,881 (3.1) | 12,137 (3.9) | 73 (3.7) |
| Ticino | 284,417 (4.3) | 2,856 (4.6) | 17,343 (5.5) | 123 (6.2) |
| Vaud | 603,119 (9.0) | 5,165 (8.4) | 34,465 (10.9) | 232 (11.8) |
| Valais | 265,481 (4.0) | 2,492 (4.0) | 7,112 (2.3) | 49 (2.5) |
| Neuchâtel | 143,272 (2.1) | 1,501 (2.4) | 9,295 (2.9) | 73 (3.7) |
| Geneva | 365,634 (5.5) | 3,013 (4.9) | 20,510 (6.5) | 112 (5.7) |
| Jura | 57,600 (0.9) | 642 (1.0) | 3,723 (1.2) | 30 (1.5) |

[a]Follow-up period defined from December 31, 2010 to December 31, 2011.

SE population, across months of death. Individuals who died in January and February were underrepresented in the SE, for example, only 4.8% of SE 2010 participants who died in 2011 did so in January, compared to 9.3% in the underlying STATPOP population. This pattern was consistent across all surveys 2010–2013. Figure S1 shows estimated MR for the STATPOP and SE population of 2010, with an increasing follow-up time until end of 2011, 2012, 2013 and 2014. We found that MR differences between the CS and IP weighting approaches diminished with increasing follow-up time.

## Simulation study

Figure S2 shows the estimated log MR of the simulated population versus the estimated weighted log MR of the simulated sample. The variability of the estimated weighted log MR of the sample was associated with the sampling fraction and the assumed percentage of deceased individuals. Unweighted analyses showed biased MR estimates for all performed scenarios. We found that IP weighted estimates were comparable to known CS weighted analyses, with a slightly higher bias in a scenario with a low sampling fraction and a low percentage of deceased individuals.

## Sensitivity analyses

Table S3 summarises crude odds ratios of being SE participant of the years 2011–2013, using covariates age, gender, nationality, civil status, and vital status. We found similar participation patterns as in the year 2010. Figure S3 shows the 1-year mortality analyses for the STATPOP and SE populations of the years 2010–2013, i.e., mortality in 2011–2014. CS weights showed a reduced bias for later sampling years, but still slightly underestimated the underlying population mortality rates.

**Table 2  Crude odds ratios (OR) of participation in the 2010 micro census, with the registry-based population 2010 as reference population.**

|  |  | OR (95% CI) |
|---|---|---|
| Age categories | [15, 20) | Reference |
|  | [20, 25) | 0.93 (0.92, 0.95) |
|  | [25, 30) | 1.02 (1.00, 1.04) |
|  | [30, 35) | 1.12 (1.10, 1.14) |
|  | [35, 40) | 1.12 (1.10, 1.14) |
|  | [40, 45) | 1.09 (1.07, 1.11) |
|  | [45, 50) | 1.09 (1.07, 1.11) |
|  | [50, 55) | 1.09 (1.07, 1.11) |
|  | [55, 60) | 1.11 (1.09, 1.13) |
|  | [60, 65) | 1.11 (1.09, 1.13) |
|  | [65, 70) | 1.15 (1.13, 1.17) |
|  | [70, 75) | 1.15 (1.12, 1.17) |
|  | [75, 80) | 1.15 (1.12, 1.17) |
|  | [80, 85) | 1.10 (1.07, 1.13) |
|  | ≥85 years | 0.91 (0.88, 0.93) |
| Gender | Men | Reference |
|  | Women | 1.04 (1.03, 1.04) |
| Nationality | Swiss | Reference |
|  | EEA | 0.94 (0.93, 0.95) |
|  | Other Europe | 0.81 (0.79, 0.82) |
|  | Other World | 0.79 (0.77, 0.81) |
| Civil status | Single | Reference |
|  | Married | 1.06 (1.05, 1.07) |
|  | Widowed | 0.99 (0.97, 1.00) |
|  | Other | 1.04 (1.03, 1.05) |
| Region | City of Zurich | 4.31 (4.24, 4.38) |
|  | Remainder of canton Zurich | Reference |
|  | City of Bern | 4.24 (4.15, 4.34) |
|  | Bernese Jura | 2.33 (2.24, 2.42) |
|  | Remainder of canton Bern | 1.07 (1.05, 1.09) |
|  | Lucerne | 2.09 (2.05, 2.13) |
|  | Uri | 1.12 (1.05, 1.19) |
|  | Schwyz | 0.98 (0.95, 1.02) |
|  | Obwalden | 0.93 (0.87, 1.00) |
|  | Nidwalden | 1.01 (0.95, 1.08) |
|  | Glarus | 0.93 (0.87, 1.00) |
|  | Zug | 1.89 (1.83, 1.95) |
|  | Fribourg | 0.96 (0.94, 0.99) |
|  | Solothurn | 1.02 (1.00, 1.05) |
|  | Basel-Stadt | 1.06 (1.03, 1.09) |
|  | Basel-Landschaft | 1.06 (1.03, 1.09) |

**Table 2** (*continued*)

|  |  | OR (95% CI) |
| --- | --- | --- |
|  | Schaffhausen | 1.02 (0.98, 1.07) |
|  | Appenzell Ausserrhoden | 1.06 (1.01, 1.12) |
|  | Appenzell Innerrhoden | 1.00 (0.90, 1.11) |
|  | St. Gallen | 1.01 (0.99, 1.03) |
|  | Graubuenden | 0.98 (0.95, 1.01) |
|  | Aargau | 2.02 (1.99, 2.06) |
|  | Thurgau | 2.01 (1.96, 2.05) |
|  | Ticino | 2.12 (2.07, 2.16) |
|  | Vaud | 1.98 (1.94, 2.01) |
|  | Valais | 0.90 (0.87, 0.92) |
|  | Neuchâtel | 2.26 (2.21, 2.32) |
|  | Geneva | 1.94 (1.90, 1.97) |
|  | Jura | 2.25 (2.17, 2.33) |
| Vital status by end of year | Alive | Reference |
|  | Death | 0.67 (0.64, 0.70) |

## DISCUSSION

We hypothesised that time to event analyses using data from the Swiss survey-based census system may biased because of informative sampling. We investigated whether an IP weighting approach leads to representative statistical estimates in the analysis of the 2010 SE sample, representing approximately 3.5% of the Swiss population aged 15 years or older at the end of 2010. The SE is a stratified random sample of the underlying population, but unweighted and CS weighted survival analyses may be biased, because the sampling is informative. We constructed IP weights, which accounted for known sampling covariates and survival information, and compared those to CS weighted MR and LE estimates. We found that CS weighted estimates were biased, whereas IP weighted estimates were comparable to those from the total Swiss population.

Unmeasured predictors, which are related to future survival and the sampling process—but which are not part of the sampling design—could cause bias in the CS weighted analysis: for example, individuals with a poor health status were less likely to respond to the SE questionnaire, but had also a higher risk of dying. Such a situation may be caused by possible survey participants nonresponse, which is related to certain population characteristics (e.g., older people), which—on the other hand—are also related to the regression outcome (see Fig. 1). Bias in the regression analysis of survey data and possible strategies to handle this kind of bias have been extensively discussed in the demographical and statistical literature (see e.g., *Pfeffermann, 2010* and the references therein). *Pizzi et al. (2012)* compared results from an internet-based cohort study sample and results from the general population and concluded that differences in estimates between the two analysis populations reflect changes in confounding pattern due to the sampling selection process. Such a change of confounding patterns is likely to influence mortality estimates in our present study using the Swiss population. Confounding patterns in our study are likely to be caused by the underrepresentation of certain subpopulations in the micro census
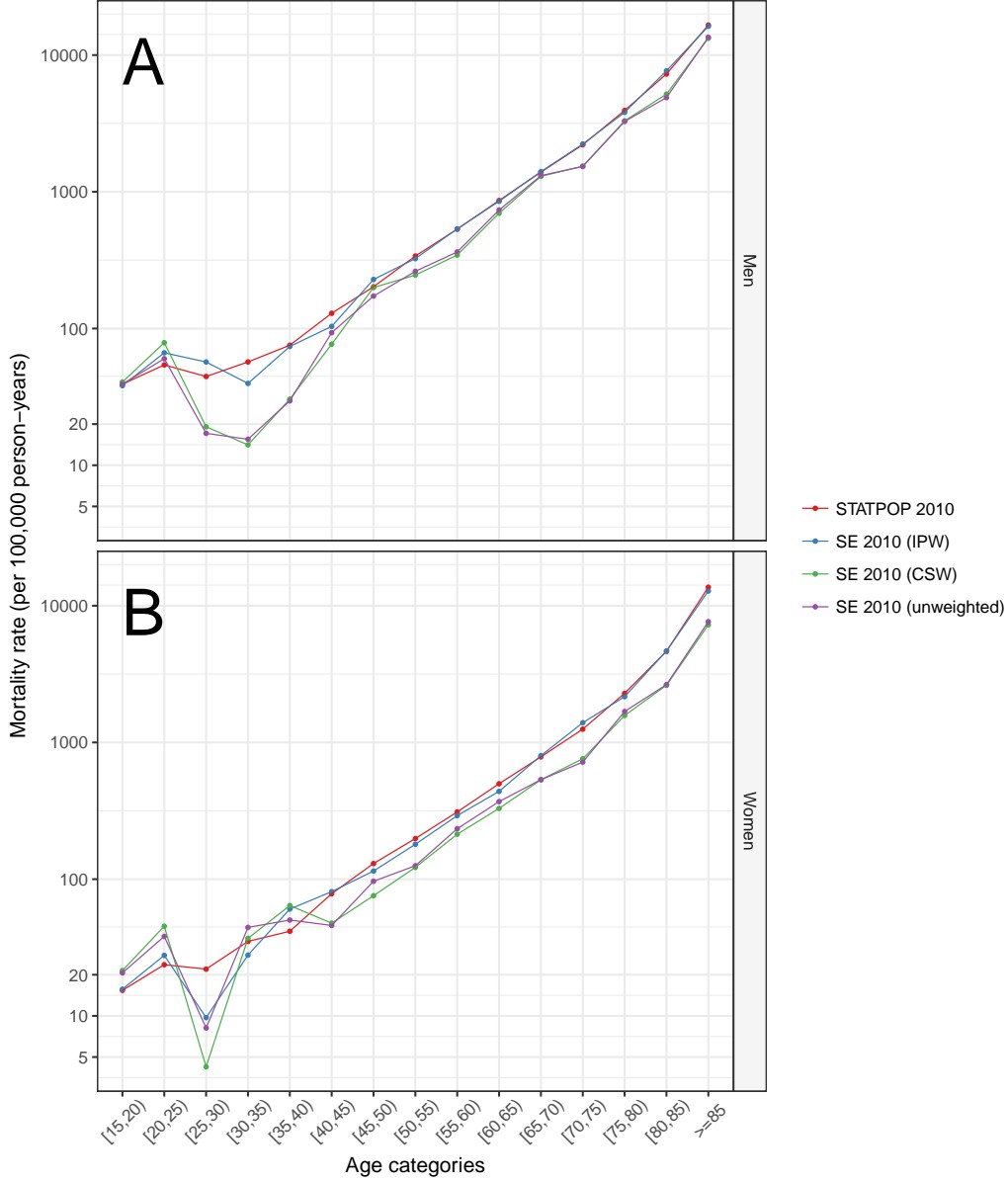

**Figure 2 Population and weighted mortality rates from December 31, 2010 to December 31, 2011, by gender\*.** (A) Men. (B) Women. Abbreviations: CSW, Calibrated survey weights; IPW, Inverse probability weights; SE, Structural enquiry; STATPOP Registry-based population. \*Permanent residents living in Switzerland aged 15 years or older at December 31, 2010. $Y$-axis is on a logarithmic scale.

sample. For example, we found that individuals who died in 2011 were less likely to having participated in the micro census of 2010. Specifically, we found that individuals who died in the early months of the year were less likely to participate in the micro census. Delayed information adjustments between the process of sending out questionnaires to individuals, and the alignment between mortality and registry information, could introduce specific nonresponse patterns. Further, our observed lower chance of micro census participation

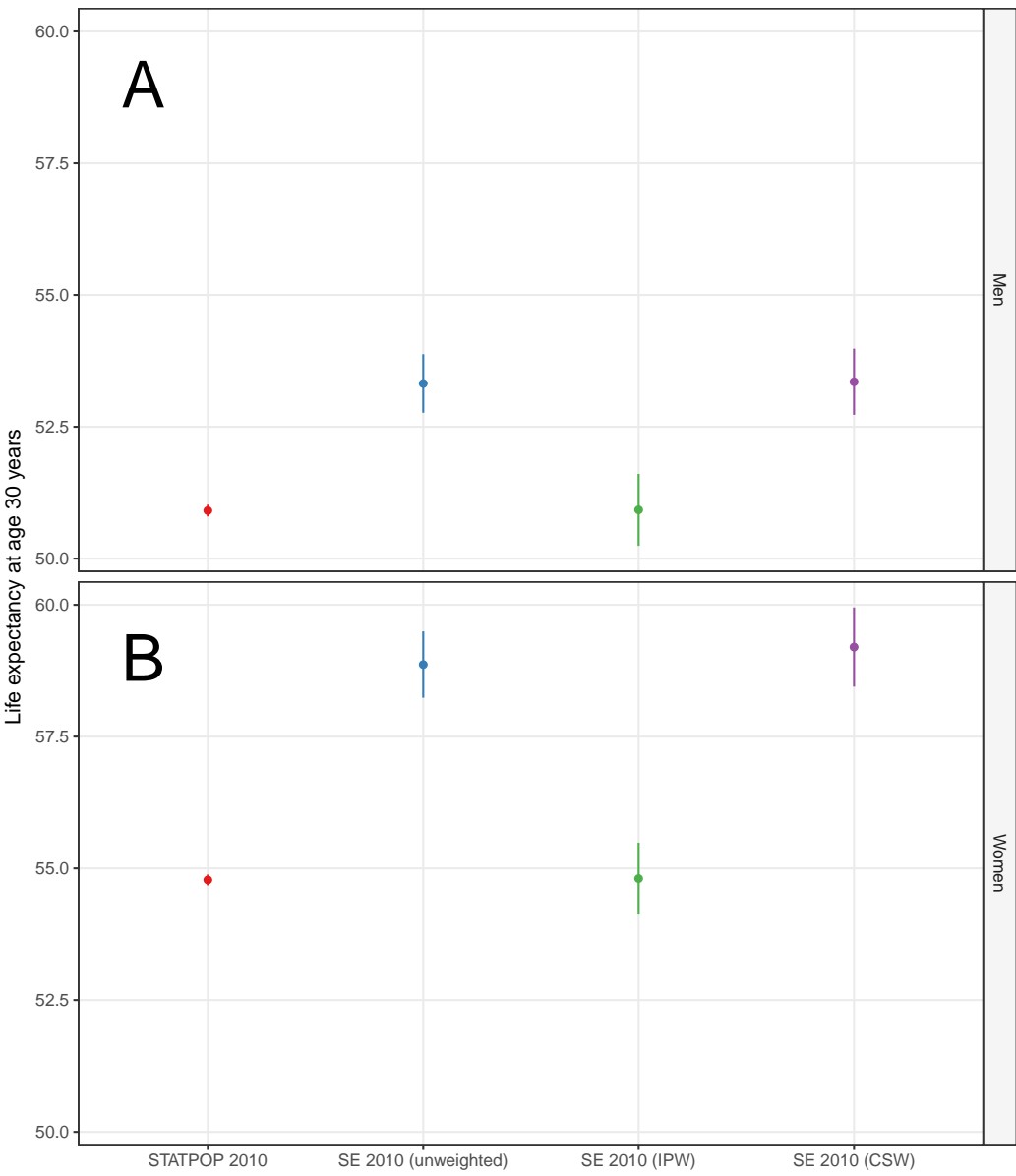

**Figure 3** **Life expectancy at age 30 years, by gender.** (A) Men. (B) Women. Abbreviations: CSW, Calibrated survey weights; IPW, Inverse probability weights; SE, Structural enquiry; STATPOP Registry-based population.

of non-Swiss individuals could be explained by a migrant effect (*Nielsen & Knardahl, 2016*). In additional analyses, we found that older foreigners tend to have a higher odd of participation in the survey, than younger individuals, which could be partly explained by the health status of these individuals (*Nielsen & Knardahl, 2016*). Such a migrant effect is likely to confound our current analysis.
When sampling design variables are related to survival time and censoring, unweighted survival estimates may be biased (*Lawless, 2003*; *Boudreau & Lawless, 2006*). However, IP weighting approaches, addressing the sampling design and the censoring information, have been shown to provide consistent population survival estimates (*Scharfstein & Robins, 2002*; *Lawless, 2003*; *Boudreau & Lawless, 2006*; *Pyy-Martikainen, 2013*). Several countries, also Switzerland, reported a higher risk of death among non-responders of surveys, compared to responders (*Barchielli & Balzi, 2002*; *Bopp, Braun & Faeh, 2014*). An Australian study investigated attrition patterns over five survey periods, starting in 1996–2008, in a population of older women, and reported that health status was associated with frailty, withdrawal from the study, and lost to follow-up (*Brilleman, Pachana & Dobson, 2010*). Such unmeasured factors, which are related to mortality and non-response, are likely leading to an informative sampling design in the new Swiss census system, and may cause bias in survival analysis of survey data. In our analyses, we found that IP weighting may adequately address the bias introduced by informative sampling. Our simulation study confirmed that IP weights, using survival information from the full population, leads to comparable MR estimates as those from known CS weights. The design of the present study—with a combination of registry-based, survey and mortality information—allowed the identification of predictors, which are associated with informative sampling, in a large population of 6.7 million individuals. This allowed us to validate our findings for consistency. Because mortality information could be deterministically linked to a registry and survey record—by using anonymous person identifiers—we could avoid uncertainty inherent to probabilistic record linkages (*Schmidlin et al., 2013*). The Swiss Harmonized Registry Act Revision of the year 2008 legally regulated the electronic exchange between communities and state personal registries, and led to virtually no lost-to-follow-up of individuals.

Our present study has limitations. First, the IP weighting approaches is a design-based approach, i.e., the probability distribution of participation in a random sample is treated as a fixed quantity (*Hansen, Madow & Tepping, 1983*; *Little, 1983*; *Little, 2004*). Design-based approaches have been shown to be only asymptotically unbiased, and potentially inefficient (*Little, 2004*; *Kim & Skinner, 2013*). In the present study we did not investigate model-based approaches, where non-participation in a random sample is predicted using regression models (*Gelman, 2007*; *Little, 2007*). Further, IP modelling relies on strong statistical assumptions which have to be fulfilled to provide unbiased statistical estimates (*Scharfstein & Robins, 2002*). In our setting, for example, exchangeability requires that the mortality among censored and uncensored individuals is the same in participating and not-participating individuals, given all predictors which predict mortality and those in the sampling design. However, researchers never know whether all relevant predictors were identified and collected in a real data setting. Second, our approach includes the outcome variable (i.e., survival status) in the IP weight construction. Thus, the validity of our estimates depends on the strength of the participation-outcome association and on the risk factor-outcome association (i.e., age and gender) (*Pizzi et al., 2011*). Third, our findings are limited to observed and available data within the registries and surveys. We could not investigate in more detail the real nonresponse patterns, i.e., which survey participants

could not be contacted from the SFSO and whether sampled persons subsequently proved to have died were handled as non-responders or if they were excluded. Detailed information on those, who were contacted, but refused to participate, would strengthen our findings and could improve our estimates. Further, in the STATPOP population we could not differentiate whether individuals lived in a collective household (i.e., nursing homes) or not. This leads to a potential numerator/denominator bias, because the SE population is restricted to individuals living in a private household.

For many decades, most European countries relied on full enumeration censuses, where information for the whole population was collected at a specific reference date. A full enumeration census approach provides very detailed information on individual level (e.g., civil status, religion, household composition), but requires considerable financial and administrative resources. Registry-based and micro-census approaches are cost-efficient and require fewer resources. Among the countries of the United Nations Economic Commission for Europe (UNECE), there was an increase of registry-based and micro census approaches from 8% of all countries in 2000, to 19% in 2010 (*UNECE (2017)*, accessed (30 November 2017)). However, from a statistical perspective they are prone to the sampling design errors, including informative sampling, and require adequate analysis techniques. In the following we give examples from other studies which analysed (or plan to analyse) mortality in samples of registry or census populations. One of the largest cohort studies worldwide—the Global Burden of Disease Study—depends on a combination of complete and partially-complete (i.e., sampling information) registry and census information from different countries worldwide (*Abajobir et al., 2017*). To account for sampling and non-response errors, the study authors differentiate between different data sources and adjust for country-specific non-sampling error in their analysis. A study from the United Kingdom analysed mortality patterns from 1991 to 2011 from three different regions and data sources (*Katikireddi et al., 2017*). The analysis was based on information from the whole population of Northern Ireland, a 5.3% sample of the Scottish population, and a 1% sample of the English and Wales population. The authors calculated mortality rates stratified by sex, by estimating the number of expected deaths for the Scottish and English/Wales samples using a simulation approach. A recent study from Finland investigated migrant mortality for the years 1939–2010 (*Haukka et al., 2017*). The authors used a 10% sample of the 1950 and 1970 Finish population censuses, and calculated all-cause and cause-specific mortality rates. The authors discuss possible selection and attrition biases in their study limitations, but did not address this kind of bias in sensitivity analyses. The German National Cohort (GNC) is a nationwide cohort study, based on a sample of 200,000 German residents, which are recruited through a network of 18 study centres (*Ahrens et al., 2014*). The authors estimated the anticipated response rate between 40 and 50%, but did not further specify how sampling bias will be included in their planned analyses. The above-mentioned examples show that a combination of registry-based information and population-sampling information play a crucial role in demographic and epidemiological research, and adequate analysis strategies are necessary to avoid bias due to non-response and non-participation.

In the present study, we concluded that in Switzerland—which uses a combination of registry and survey-based censuses—unweighted or CS weighted analyses lead to biased estimates, if the main outcome is survival time. The observed bias from unweighted and CS weighted survival analysis is of importance for absolute quantities (for example, mortality rates or life expectancy), but the bias may be less pronounced in relative outcomes measures (for example, hazard ratios or odds ratios). We highlight that the IP approach is not only restricted to the described setting of a change from a full enumeration census to a micro census system, but may be generalized to other settings. For example, non-response in a full enumeration census or lost-to-follow-up scenarios can be addressed by this approach to improve longitudinal validity, given the non-response or non-participation is known in the study population. The underrepresentation of population subgroups in survey data and possible selection bias have been extensively discussed in the literature, but survey-based mortality analyses often lack validation using registry data, or external data in general. *Keiding & Louis (2016)* exemplary discuss the limited generalisability of survey data when external validation is lacking for internet-based surveys with possible self-selection. Obviously, generalisability is more limited if the survey participation is informative.

## CONCLUSION

We conclude from our study that weighted regression analysis of census-linked survey data has to be conducted carefully, and if possible, validated by registry-population information. If a validation or a registry sampling weight construction is missing, estimation results have to be interpreted with caution and might be biased.

## ACKNOWLEDGEMENTS

We thank the Swiss Federal Statistical Office for providing mortality and census data and for the support which made the Swiss National Cohort and this study possible. The members of the Swiss National Cohort Study Group are Matthias Egger (Chairman of the Executive Board), Adrian Spoerri and Marcel Zwahlen (all Bern), Milo Puhan (Chairman of the Scientific Board), Matthias Bopp (both Zurich), Nino Künzli, Martin Röösli (both Basel), Murielle Bochud (Lausanne) and Michel Oris (Geneva). Further, we thank Veronika Skrivankova for helpful comments in the preparation of the manuscript.

### Funding

The Swiss National Cohort was supported by the Swiss National Science Foundation (grant nos. 3347CO-108806, 33CS30_134273 and 33CS30_148415). The funders had no role in study design, data collection and analysis, decision to publish, or preparation of the manuscript.

### Grant Disclosures

The following grant information was disclosed by the authors:
Swiss National Science Foundation: 3347CO-108806, 33CS30_134273, 33CS30_148415.

## Competing Interests

The authors declare there are no competing interests.

## Author Contributions

- André Moser conceived and designed the experiments, performed the experiments, analyzed the data, contributed reagents/materials/analysis tools, wrote the paper, prepared figures and/or tables, reviewed drafts of the paper.
- Matthias Bopp conceived and designed the experiments, performed the experiments, contributed reagents/materials/analysis tools, wrote the paper, reviewed drafts of the paper.
- Marcel Zwahlen conceived and designed the experiments, performed the experiments, analyzed the data, contributed reagents/materials/analysis tools, wrote the paper, reviewed drafts of the paper.

## Human Ethics

The following information was supplied relating to ethical approvals (i.e., approving body and any reference numbers):

Federal laws gave the federal authorities the right to collect (without consent) the census and mortality data used in the SNC. Approval for the anonymous linkage in the SNC was obtained from the Ethics Committees of the Cantons of Zürich (approval no. 13/06) and Bern (approval no. 205/06).

## Data Availability

Moser, André; Bopp, Matthias; Zwahlen, Marcel (2018). Calibration adjustments to address bias in mortality analyses due to informative sampling—a census-linked survey analysis in Switzerland: https://boris.unibe.ch/109954/.

DOI: 10.7892/boris.109954.

## Supplemental Information

Supplemental information for this article can be found online at http://dx.doi.org/10.7717/peerj.4376#supplemental-information.

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
