# Peer review of "Calibration adjustments to address bias in mortality analyses due to informative sampling—a census-linked survey analysis in Switzerland"

_PeerJ, doi:10.7717/peerj.4376_

## Round 0.1 · original submission · Minor Revisions

I second the reviewers' comments that minor English editing is needed in places, and that it would be useful to know more about what other countries have adopted or are considering adopting this approach (the reviewers and editor are from the U.S. and Canada).

Thank you for submitting your work to PeerJ.

·

Basic reporting

Clear and unambiguous english is used for the most part making the manuscript comprehensible. Some editorial revisions including the suggestions to the author below would improve clarity.

Experimental design

The authors effectively demonstrate how a survey-based census system without consideration for differential non-response, such as that adopted in Switzerland, will impact the validity of survival and/or mortality analyses. They do so by comparing mortality rates and life expectancy derived from survey weights with those derived from inverse probability weights using survival data from the country’s Vital Stats Registry and taking into consideration survey non-response data. The analysis is adequate and understandable, with appropriate tables and figures, provided table 1 is modified. The findings and conclusions are accurate and of importance.

Validity of the findings

While the overall findings are valid and important and the conclusion is well stated, the general context of this research question is not discussed especially the practical implications of these findings (IMPACT). Moreover the baseline data of STATPOP could be better described. For the rest, the data seems robust , statistically sound and controlled.

Additional comments

I read the manuscript by Moser et al. with great interest. The authors effectively demonstrate how a survey-based census system without consideration for differential non-response, such as that adopted in Switzerland, will impact the validity of survival and/or mortality analyses. They do so by comparing mortality rates and life expectancy derived from survey weights with those derived from inverse probability weights using survival data from the country’s Vital Stats Registry and taking into consideration survey non-response data. The analysis is adequate and understandable, with clear, appropriate tables and figures. The findings and conclusions are accurate and of importance. Some concerns are as follows: Major concerns:

• While the manuscript shows that estimates based on survey weights will be underestimated (MR) and overestimated (LE), it falls short of explaining the impact or significance of this concern. For population-based studies on vital statistics, one could always fall back on the STATPOP file; thus, it is important for the authors to convey what the consequences of this difference in estimates might be in practice, ideally providing concrete examples.
• It would be advantageous to provide the regional European context. Is this a local problem, or are other countries reverting to the same census plans?
• Survival Concerns: Please discuss the completeness of mortality records for linkage with STATPOP. Regarding the mortality records of non-Swiss populations (roughly 20% according to Table 1), what is the “lost to follow-up” rate? How is vital status ascertained? (death certificate alone? Regular collection of “state income or pension” anywhere in the world? Etc.) What about people moving to warmer geographies or to their countries of origin and dying there (some with the authors’ exact characteristic of interest - poor health status)? How would these aspects affect the survival used for the IP weights? Furthermore, due to the overlap with poor health status, how would it affect the estimates of the current study? At minimum, these concerns could be addressed in the limitations section.
• Non-Swiss populations have significantly lower odds of response to the survey, but the effect of that was not discussed. Please also clarify if cross-border workers to cities like Geneva and Basel are included in STATPOP –as part of the Swiss population - and/or the survey?

Minor suggestions:

• The word “informative” may come across as positive to the less informed reader. The text should be modified so that “informative” is clearly portrayed as an unwanted characteristic.
• Please check the order of the Figures. Currently starts at 3.
• Lines 129-131: Is the survival time individually updated as per some sort of contact with the subject or is survival time measured throughout the full period presuming the subject is alive in the absence of a death record?
• Line 144: Does the OR=4 come from Pizzi et al. as well? If not, please substantiate your choice.
• Line 149: The sentence “The overall percentage of deaths was set to 10% and 25%” needs to be clarified with information regarding which group or data subset is involved.
• Lines 262-263: This sentence is best suited for the Introduction.
• Lines 303-305: This statement is an overreach and possibly wishful thinking. Readers will understand that resource shortages are driving countries towards novel approaches to counting their populations. Yet, as researchers, we know that there is no real substitute for a Full Enumeration Census (albeit also imperfect). To suggest that surveys and statistical estimates could ever replace an actual total count is bold at the least, and surely contentious, unless Switzerland is unique and there is a total absence of undocumented persons, the “lost-to-follow-up” rate is zero, and the borders are fully controlled. I would strongly consider modifying this segment of text.

Minor language/word choice concerns:

While the manuscript is comprehensible, some editorial revisions including the following would improve clarity.
1-The use of the word “register” is awkward
2-The use of the word “consistent” -it often seems like the authors actually mean accurate or valid
3- “Despite” is not used correctly; sentences using it should be modified.
4- The use of future tense in some sentences is odd/unusual.
5- Other awkward phrases:
 true underlying population estimates (need a better word for “true”)
 they address for the sampling
 because of privacy preservation
 a bad health status
 studies including Switzerland
 state-of-the-art nowadays
 provided almost
In addition, table 1 is confusing and should be modified, possibly with horizontal lines.

Reviewer 2 ·

Basic reporting

1. Throughout the article, the switch from a full census to a survey-based micro-census is provided as justification for the work. Presumably, however, full censuses also have non-response and the same conclusions could apply (i.e. inverse probability weights would still improve the longitudinal validity, despite their not being a sampling probability). Please clarify the generalizability of your findings, and if they are generalizable beyond this micro-census scenario, perhaps use a different motivation to justify your study.
2. Abstract: Define a registry-based census. Be consistent with calling it either a registry-based or register-based census.
3. Line 35: Describe who the 6.5 million individuals are (i.e., all persons included in the registry-based census)
4. Line 37: Clarify “aimed to be sampled”. Presumably 3.5% were sampled, but not all responded? Perhaps remove “aimed to”
5. Line 74: “the statistical analyses” is vague – what statistical analyses?
6. Lines 80/81: This sentence is very general and therefore confusing. I would differentiate between what your aim was (specific, mortality rates) and its implication (potentially broader, any time to event data)
7. Line 94: Remove “will”
8. Lines 111, 121: English language editing needed.
9. Line 136: Figures should be numbered in the order they appear in the text
10. Lines 140, 242, Figure 3: Replace “a bad health” with “poor health status” throughout.
11. Line 185, Table 1: I may be ignorant given that this is not my area of expertise, but are the mean CS and IP weights informative to the reader? I am not sure how to interpret these in Table 1 and wonder whether there is use in them being there?
12. Figure 2: Suggest making the x-axis categorical with the different methods across the x-axis. Could also consolidate to one graph.

Experimental design

1. The simulation study seems like a bit of an afterthought, both in the presentation and the interpretation of the results. I suggest either fully justifying the usefulness and implications of the simulation study, or leaving it out altogether.

Validity of the findings

No comment.

Additional comments

Overall, I think this is a well-executed and thorough study that is well described and has valid findings. Some English language editing is needed and some further clarification of the generalizability and implications of the results would improve the paper, but on the whole it seems to be a quality piece of work worthy of publication.

---

## Round 0.2 · accepted · Accept

You have adequately addressed each of the reviewers' comments and the paper is improved as a result.